

# Mapping the vertical heterogeneity of Greenland's firn from 2011-2019 using airborne radar and laser altimetry

Anja Rutishauser[1], Kirk M. Scanlan[2], Baptiste Vandecrux[1], Nanna B. Karlsson[1], Nicolas Jullien[3], Andreas P. Ahlstrøm[1], Robert S. Fausto[1], Penelope How[1]

[1]Department of Glaciology and Climate, Geological Survey of Denmark and Greenland, Copenhagen, Denmark
[2]DTU Space, Technical University of Denmark, Kgs. Lyngby, Denmark
[3]Department of Geosciences, University of Fribourg, Fribourg, Switzerland

*Correspondence to*: Anja Rutishauser (aru@geus.dk)

**Abstract.** The firn layer on the Greenland Ice Sheet (GrIS) plays a crucial role in buffering surface meltwater runoff, which is constrained by the available firn pore space and impermeable ice layers that limit deeper meltwater percolation. Understanding these firn properties is essential for predicting current and future meltwater runoff and its contribution to global sea-level rise. While very high-frequency (VHF) radars have been extensively used for surveying the GrIS, their

lower bandwidth restricts direct firn stratigraphy extraction. In this study, we use concurrent VHF airborne radar and laser altimetry data collected as part of Operation Ice Bridge (OIB) over the period 2011-2019 to investigate vertical offsets in the radar surface reflection ($dz$). Our results, corroborated by modelling and firn core analyses, show that a $dz$ larger than 1 m is strongly related to the vertical heterogeneity of a firn profile, and effectively delineates between vertically homogeneous and vertically heterogeneous firn profiles. Temporal variations in $dz$ align with climatic events and reveal an expansion of

heterogeneous firn between 2011-2013 covering an area of ~338,450 km$^2$, followed by firn replenishment over the years 2014-2019 spanning an area of ~664,734 km$^2$. Our approach reveals the firn evolution of key regions on the Greenland Ice Sheet, providing valuable insights for detecting potential alterations in meltwater runoff patterns.

## 1   Introduction

The Greenland Ice Sheet (GrIS) is a major contributor to global sea level rise, where surface runoff currently accounts for

about 50% of the total GrIS mass loss (The IMBIE Team et al., 2020). Firn (compacted snow that has survived at least one melting season) is a key component in the GrIS surface mass balance, and currently retains between 41-46 % of the surface meltwater (Pfeffer et al., 1991; Harper et al., 2012; Steger et al., 2017). This retention acts as a significant buffer against meltwater runoff and, consequently, sea-level rise. However, impermeable ice slabs and densification of firn restricts meltwater percolation and storage; thereby increasing surface runoff (Machguth et al., 2016; MacFerrin et al., 2019). These

ice slabs typically form through the accretion of ice between-, at the top-, or bottom of pre-existing thin ice layers in the firn (Jullien et al., 2023; MacFerrin et al., 2019). Understanding firn properties and the extent of thin ice layers in firn is crucial for assessing current and future GrIS surface mass balance, and important for interpreting satellite radar altimetry



measurements, which can have ambiguities due to complex near-surface firn stratigraphy (Nilsson et al., 2015). Finally, knowledge of firn density is also required when converting surface height changes from repeat altimetry to mass balance

estimates (Sørensen et al., 2011).

Research on firn properties and their impact on meltwater retention and surface mass balance can generally be grouped into three categories: First, detailed in-situ measurements at selected sites, focusing on aspects like firn air content and melt features in firn cores (Rennermalm et al., 2021; Machguth et al., 2016; Kameda et al., 1995; Braithwaite et al., 1994),

meltwater percolation monitoring via subsurface temperature measurements (Humphrey et al., 2012), time domain reflectometry (Samimi et al., 2021), upward-looking ground-penetrating radar measurements (Heilig et al., 2018) or firn stratigraphy from local-scale GPR surveys (Brown et al., 2011). Second, airborne or space-borne remote sensing techniques have been used to map spatially extensive firn properties. For instance, the ultra-high frequency (UHF) Accumulation Radar (AR) deployed during Operation Ice Bridge (OIB) airborne surveys over the GrIS has been used to directly map ice layers

(Culberg et al., 2021), ice slabs (Jullien et al., 2023; MacFerrin et al., 2019), firn aquifers (Forster et al., 2014; Miège et al., 2016; Horlings et al., 2022) and retrieve past accumulation rates (Karlsson et al., 2016; Lewis et al., 2017). Extensive radar surveys have also been collected with very high frequency (VHF) radars over the GrIS, however, these are typically deployed to retrieve ice thickness and englacial layers, whereas their lower bandwidth prevents them from directly resolving the firn stratigraphy. Nonetheless, VHF datasets over Antarctica and the Canadian Arctic have been used to characterize firn

properties through statistical analysis of the surface echo strengths (Grima et al., 2014; Rutishauser et al., 2016; Chan et al., 2023). Recent work demonstrated the use of passive microwave satellite measurements to map firn facies and liquid water in firn (Miller et al., 2022a, b; Colliander et al., 2023), while multi-frequency satellite radar can also be used to estimate the firn density and structure (Scanlan et al., 2023). Satellite optical observations have been used to track runoff from the firn area (Tedstone and Machguth, 2022). Lastly, physical or statistical firn models (Langen et al., 2017; Brils et al., 2022; Medley et

al., 2022; Thompson-Munson et al., 2023; Vandecrux et al., 2019) are used to bridge the spatial gap between localised measurements. However, statistical models are limited by data availability and physical models are still unable to capture the complex processes taking place in the firn of ice sheets (Lundin et al., 2017; Vandecrux et al., 2020).

Here, we leverage OIB measurements, specifically the VHF airborne radar (Multichannel Coherent Radar Depth Sounder,

MCoRDS) and laser altimetry datasets, to investigate firn properties across the GrIS in the period 2011-2019 (Figure 1), and to provide insights into how GrIS-wide firn conditions evolve over time. Our approach is based on the hypothesis that a vertically heterogeneous firn structure, resulting from repeated melting and refreezing events, affects the airborne VHF radar signal such that the returned surface reflection appears below the actual ice sheet surface. We first test this hypothesis through numerical modelling of the radar signal, examining how near-surface density heterogeneities impact the MCoRDS

surface reflections. We then derive GrIS-wide radar surface reflection offsets via a comparison of the MCoRDS-derived ice sheet surface height to those measured with the Airborne Topographic Mapper (ATM) laser altimeter, assuming that the



laser altimeter reveals the true ice sheet surface elevation. Finally, we compare the observed surface return offsets to in-situ firn measurements, and together with the modelling results, classify zones of homogeneous and heterogeneous firn and track their evolution over the period 2011-2019.

## 2    Data and methods

### 2.1    Radar sounding and laser altimetry data

We use data from the Multichannel Coherent Radar Depth Sounder (MCoRDS, versions 2 and 3) and Airborne Topographic Mapper (ATM) laser altimeter, both collected during NASA's OIB surveys over the GrIS between 2011-2019 (Figure 1). The MCoRDS radar operates at a centre frequency of 195 MHz and has a 30 MHz bandwidth. The radar's surface return signal is affected by dielectric contrasts in the snow, firn and ice within the depth volume determined by the theoretical vertical range resolution $\delta_r$, given as

$$\delta_r = \frac{ck_t}{2B\sqrt{\varepsilon_r}}, \tag{1}$$

where c is the speed of light in vacuum, $k_t$ is a windowing factor, B is the bandwidth and $\varepsilon_r$ is the dielectric permittivity of the probed subsurface. For MCoRDS ($k_t = 2$ (Rodriguez-Morales et al., 2014)), the theoretical vertical range resolution is 5.6 m in ice ($\varepsilon_r = 3.15$) and 7.5 m in firn ($\varepsilon_r = 1.8$, corresponding to a firn density of 384.4 kg/m³). The MCoRDS radar data have a nominal footprint at the ice sheet surface ($Dpl = 2\sqrt{\frac{ck_t}{B}}R$, where R is the aircraft height above the ice sheet) of ~200 m and an along-track trace spacing of 14-30 m, depending on the survey year (see Table S1). We extract the radar surface heights from the *csarp_standard* low-gain data product (Data_img_01), except for the 2011 survey year, where only the *csarp_combined* data was available.

The ATM laser altimetry data are collected simultaneously with the MCoRDS data. We utilise the ATM Level 2 product, which has been resampled along the tracks and contains 3-5 platelets spanning the 80 m cross-track swath of the ATM scan (Studinger, 2014).



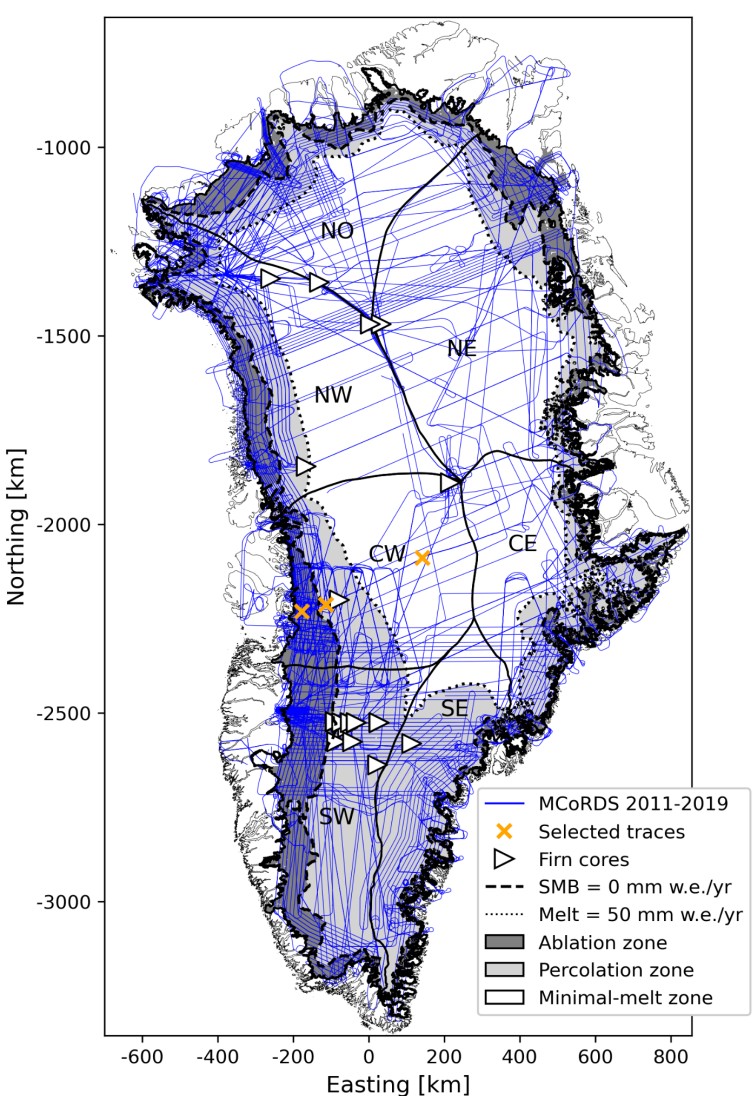

**Figure 1: Map showing the MCoRDS radar profiles between 2011-2019 used to derive surface peak offsets, along with the SUMup**
**firn cores from overlapping survey years and the location of selected example traces shown in Figure 2. Mean annual Modèle Atmosphérique Régional (MAR) surface mass balance (SMB) and melt estimates between 2009-2019 (Fettweis et al., 2017) are used to outline the ablation zone (SMB<0), the percolation zone (SMB≥0 and melt ≥ 50 mm w.e./yr) and a minimal-melt zone (melt < 50 mm w.e./yr), which we hereby refer to as the dry-snow zone. Catchment basins are derived from Mouginot and Rignot (2019).**

## 2.2 Picking of the MCoRDS ice sheet surface reflection

Figure 2 presents examples of MCoRDS surface returns (hereby referred to as the broader surface signal of elevated amplitudes, encompassing multiple peaks) of different glacier facies in Central West Greenland (locations shown in Figure 1). Generally, the surface returns from the ablation- and dry-snow zones display distinct, narrow single peaks. In contrast, those from the percolation zone feature broader, and often multi-peaked returns, complicating the identification of the ice



sheet surface. While the Center for Remote Sensing and Integrated Systems (CReSIS) provides ice-sheet surface picks along

with the radar data, we find that these often correspond to the maximum peak amplitude. However, in instances where a

smaller peak or a "bulge" precedes the maximum peak, that first peak likely represents the air-ice sheet interface, with

subsequent peaks arising from internal firn density contrasts, such as ice layers (see Section 3).

To consistently identify the reflection most representative of the air-ice sheet interface, we apply a custom re-picking

algorithm. This algorithm operates on a trace-by-trace basis and involves the following steps: 1) extracting a 40-sample (1.3

µs) vertical window centred around the CReSIS provided surface pick, 2) up-sampling the signal by a factor of 10 via

padding of the signal in the frequency domain to enable sub-sample peak location estimates and the identification of

"bulges" that often precede the maximum peak (e.g. Figure 2c), 3) normalising the trace signal amplitudes between 0 to 1,

and 4) selecting the first peak in the up-sampled signal that exceeds a 0.1 amplitude threshold (10% above the noise floor) in

both the up-sampled and original signals. Applying the 10% threshold on the original signal minimises the risk of picking

numerical artefacts introduced by up-sampling, while allowing to identify bulges that do not appear as peaks in the original

MCoRDS signal (e.g. Figure 2c).

Visual inspections of selected radar profiles confirms that the re-picked surface reflections are reasonable. A vertical picking

error of 30 ns (approximately one fast-time sample) would translate to a surface height offset between 2.5 m (ice) to 3.4 m

(firn). To mitigate the impact of outliers arising from potentially misidentified surface peaks, we apply a moving average

window over the mean radar footprint diameter (calculated for each profile) to the repicked ice sheet surface height.

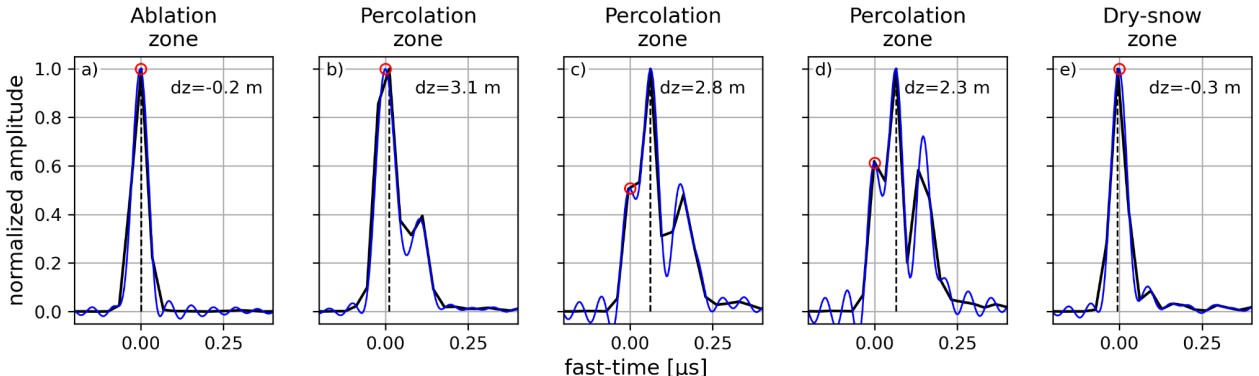

**Figure 2: Example MCoRDS radar traces along Profile A (Figure 7) collected in 2017 (see Figure 1 for trace location) showing**
**typical surface returns over the ablation zone (a), percolation zone (b-d, note the three traces are neighbouring traces and**
**represented with one location marker on Figure 1), and dry snow zone (e). Black is the measured MCoRDS signal and blue is the**
**up-sampled signal. The black dashed line shows the peak provided by CReSIS whereas the red dot is the re-picked reflection used**
**as ice sheet surface representative and to calculate the radar peak offsets. The trace energy is normalised, and fast-times**
**(traveltime) are presented relative to the re-picked surface.**



### 2.3 Derivation and calibration of radar surface peak offsets

To calculate the vertical offsets ($dz$) between the radar-derived ice sheet surface ($h_{radar}$) and the actual ice sheet surface as measured by laser altimetry ($h_{laser}$), we use:

$$dz = h_{laser} - h_{radar} \, .$$

To ensure compatibility, $h_{laser}$ is calculated as the mean of all laser observations within the radar's pulse-limited footprint.

Once the surface peak offset ($dz$) is calculated for each radar observation, we apply several exclusion criteria to ensure the best possible quality of the $dz$ dataset: The MEaSUREs Greenland Ice Mapping Project (GIMP, 90 m resolution) ice mask (Howat et al., 2014) is used to remove data points outside ice-covered areas. Data points where the aircraft roll angle exceeded 0.1° are omitted to ensure nadir-pointing of both instruments. For the 2011 and 2012 datasets lacking roll angle information, we use the deviation of 10 km long flight line segments from a straight line as a roll angle proxy. Locations with fewer than 10 laser points within the radar footprint are also excluded, as are data points where the aircraft's range to the ice sheet surface is below 400 m (due to interference/cut-off of the ice sheet surface reflection). Finally, a few poor-quality profiles were identified through visual inspection of the radargrams and the derived peak offsets (see Table S1). After all filtering, approximately 15.5 million $dz$ observations for the period 20111-2019 remain.

The modelling results (see Section 3) suggest that radar- and laser-derived surfaces should align ($dz = 0$) in the ablation zone, where the primary dielectric contrast is the air-ice transition. However, we observed systematic offsets between $h_{laser}$ and $h_{radar}$ even over the ablation zone in some years and survey days. These could be caused by uncorrected timing issues (i.e. cable delays) in the radar measurements. To ensure consistency between measurements from different seasons, we define a calibration area in the ablation zone with ample data from all seasons (Figure S1), starting 20 km East from the western ice margin to avoid the steepest and most crevassed part of the ice, and extending to the maximum end-of-summer snowline elevation between 2009-2018 (Fausto et al., 2018). For each season, and specific survey days when necessary, we adjust all data by subtracting the median $dz$ calculated over the calibration area (Table S1).

After calibration, we remove physically unreasonable $dz$ values that fall outside the 2-98 percentile range for each survey year, as these likely result from incorrect surface reflection picks. Finally, to analyse large-scale trends, $dz$ data points are averaged onto a 10x10 km grid over the GrIS, where grid gaps smaller than 30 km are filled using linear interpolation.

### 2.4 Numerical modelling of the radar surface reflection

The radar return from a simple two-layer medium, such as the atmosphere over bare ice, typically consists of a single peak, with the peak centred at the interface between the two layers. However, over a medium with multiple interfaces within the radar range resolution depth - like firn containing several ice layers - the radar surface reflection is more complex. To investigate how firn properties affect the MCoRDS radar surface return, we employ RadSPy (https://github.com/scourvil/RadSPy), a radar sounding simulator in Python (Courville and Perry, 2021). RadSPy is an open source, one-dimensional electromagnetic wave forward-modelling software originally developed to simulate radar



reflections observed with the Mars Reconnaissance Orbiter's Shallow Radar (SHARAD) (Courville and Perry, 2021). RadSPy assumes that the N-layered input model consists of flat layers without any roughness, a normally incident plane wave, and no dispersion of the radar signal through the media. For simulating the MCoRDS waveform, we use a one-
160    microsecond long chirp swept over the 180-210 MHz MCoRDS frequency range following Equation 15 in Courville and Perry (2021).

We employ various input models to represent different glacier profiles as summarised in



Table 1. For the ablation zone, the model domain comprises two half-spaces; atmosphere and ice. Here, we also explore the impact of adding a snow layer (three-layer model) of varying thickness (up to 2 m) and a density of 341 kg/m³, corresponding to the mean density measured in the top 0.5 m across the GrIS (Fausto et al., 2018). In the dry-snow zone, we use a multi-layer model consisting of the atmosphere above an idealised ice-free firn density-depth profile ($\rho_{firn} = 320.6\ kg/m^3 + 15.4d$, Figure S2), which is constructed from in situ firn cores (see Section 2.5). The input profile representing the firn layer is segmented into 5 cm thick layers, where each layer contains the bulk density for the given depth range. For the percolation zone, we introduce single or multiple (evenly spaced) ice layers (density of 862 kg/m³ (Rennermalm et al., 2021)) at varying depths and thicknesses into the ice-free density profile.

The input density profiles (ρ in kg/m³) are converted to dielectric permittivity (ε) profiles using the empirical model by Kovacs et al. (1995):

$$\varepsilon = (1 + 0.845 * 10^{-3}\rho)^2 \tag{2}$$

The RadSPy-modelled signals are normalised to their maximum amplitude, and the ice sheet surface reflection (representing the atmosphere-ice/snow/firn interface) is identified as the first peak exceeding a threshold of 10% of the maximum amplitude, similar to our approach with MCoRDS data (see Section 2.2). We note that while we expect changes in reflection amplitude for different firn profiles, here we only focus on the vertical (fast-time) position of the surface peak and not the power itself.

Finally, the RadSPy modeled signal has a higher sampling rate (1.2 ns) than the MCoRDS signal (33.3 ns). To assess the impacts of waveform sampling on the resultant peak offsets, we also examine how downsampling the modeled waveform influences the vertical position of the surface reflection peak.



**Table 1: Summary of the different modelling experiments representative of different glacier profiles.**

| Representative glacier profiles | Input models |
|---|---|
| Ablation zone | Two-layer model: [atmosphere[a]] [ice[b]] |
| | Three-layer model: [atmosphere[a]] [snow[c]] [ice[b]]<br>(snow thickness varying between 0.1-2 m) |
| Percolation zone | Multi-layer model: [atmosphere[a]] [firn[d] with a single ice layer[e]]<br>(varying ice layer thickness and depth) |
| | Multi-layer model: [atmosphere[a]] [firn[d] with multiple ice layers[e]]<br>(varying ice layer thicknesses, layers evenly spaced over 0.5-7 m depth) |
| Dry-snow zone | Multi-layer model: [atmosphere[a]] [firn[d]] |

Dielectric permittivities and densities used are: [a]$\varepsilon_{r\_air}$ = 1, [b]$\varepsilon_{r\_ice}$ = 3.15, [c]$\varepsilon_{r\_snow}$ = 1.7 (corresponding to 341 kg/m³ (Fausto et al., 2018), [d]firn is an idealized ice-free density-depth profile following ($\rho_{firn} = 320.6 \ kg/m^3 + 15.4d$ (Figure S2)), [e]$\varepsilon_{r\_ice\_layer}$ = 2.99 (corresponding to 862 kg/m³ (Rennermalm et al., 2021)).


## 2.5    SUMup firn cores

We use firn core density and stratigraphy from the Surface Mass Balance and Snow Depth on Sea Ice Working Group (SUMup) snow density subdataset, Greenland and Antarctica, 1952-2019 (Montgomery et al., 2018; Thompson-Munson et al., 2022).

We first use 401 SUMup firn cores collected between 2011-2019 to create a realistic ice-free firn density profile (Figure S2). This profile also serves as a baseline to which ice layers can be added for simulating different firn scenarios in our waveform modelling. The ice-free firn density profile is derived for the top 10 m by fitting a linear regression to density values below 750 kg/m³, a threshold well below the typical ice layer density of 862 kg/m³ (Rennermalm et al., 2021).

Secondly, we evaluate the MCoRDS-derived surface peak offsets $dz$ with 28 SUMup cores (Figure 1) that are: i) collected
within 5 km and the same year as the radar measurements; and ii) at least 7.5 m deep, which is the theoretical depth sensitivity of the radar surface reflection.

## 3    Simulated imprints of ice layers on the radar surface reflection

Figure 3 presents RadSPy-simulated waveforms alongside observed MCoRDS radar traces (same as in Figure 2a, b, e) for density profiles typical of the ablation-, percolation- and dry-snow zones. For both the ablation (Figure 3b) and dry-snow
zones (Figure 3d), the simulated surface signals exhibit a narrow surface return peak (0.15 µs peak width) at 0 µs. This aligns with the expected timing of the surface echo and confirms that, in the absence of subsurface density variations, the



radar surface reflection accurately represents the air-firn or air-ice interface ($dz$=0). These findings are also in agreement with earlier studies that focused on the returned power of surface reflections (Grima et al., 2014).

Introducing a snow layer into the ablation zone model (Figure 3e) generally has a minimal effect on the peak offset $dz$ for most snow depths below 1 m, but shows a trend of increasing $dz$ with increasing snow depth (Figure 3f). However, specific snow layer thicknesses - approximately 0.3 m, 0.9 m and 1.5 m - introduce peak offsets on the order of ±5 m. These thicknesses correspond to odd multiples of a quarter MCoRDS wavelength in snow (wavelength of ~1.2 m), suggesting that the observed peak offsets result from constructive interference between the air-snow and snow-ice interfaces. Indeed, the

modelled waveforms reveal a double peak in the surface return, or a bulge preceding the main peak around these particular snow layer thicknesses (Figure 3g). Here, the primary peak (or bulge) likely corresponds to the air-snow interface, and the secondary peak is linked to the snow-ice interface. This waveform pattern affects the vertical positioning of the peak identified as the surface reflection, such that the picked reflection is shifted upwards (negative fast-time/$dz$) when the returned waveform displays two distinct peaks, and downwards (positive time/$dz$) when there is a singular, but broadened

peak.





**Figure 3: RadSPy model input depth-density profiles, representing the ablation- (blue), percolation- (teal), and dry-snow zones (brown). b-c) RadSPy modelled signals for the different input stratigraphies. The coloured curves represent the modelled signal,**
**while the black curves represent typical MCoRDS traces observed over the different firn facies (Figure 2). e) RadSPy model input depth-density profile representing a snow layer on top of the ablation zone. f) Modelled surface peak offset over the ablation zone as function of the snow layer thickness. g) examples of modelled waveforms for selected snow layer thicknesses (marked with crosses on f).**

For density profiles representative of the percolation zone, the vertical position (i.e. fast-time) of the surface reflection peak

varies significantly, with *dz* values ranging from -6.5 m to 13.5 m (Figure 4 and Figure 5). Figure 4 shows how *dz* is

influenced by the changing depth and thickness of a single ice layer. While *dz* generally increases with increasing ice layer

depth and thickness (Figure 4a, d and g), it also oscillates at regular intervals, indicating a complex and non-linear

relationship between these parameters and *dz*. Inspecting individual waveforms (Figure 4e,f,i, and h) shows that the distinct

negative and positive *dz* values are associated with the presence of a double peak (negative fast-time/*dz*) and bulge (positive

fast-time/*dz*) preceding the main peak, respectively. Similar to the case for a snow layer in the ablation zone, the primary

bulge/peak is likely associated with the air-firn transition, and the secondary peak with the firn-ice layer interface. Here, the

ice layer itself presents two interfaces (top and bottom) which both contribute to the surface return signal.





When an ice layer lies at depths greater than ~8 m, the radar's surface return displays two distinct peaks. In this scenario, the secondary peak - attributed to the interface between the firn and the ice layer - no longer influences the primary peak, which

represents the air-firn transition (light teal waveform in Figure 4f). Thus, ice layers located deeper than ~8 m have negligible impact on the surface reflection peaks, leading to $dz \approx 0$. Similarly, if an ice layer is sufficiently thick and positioned such that the lower boundary is deeper than ~7.5 m deep, the bottom interface of the ice layer does not affect the surface signal (light teal waveform in Figure 4i). However, in such cases, the surface reflection peak is still influenced by the upper firn-ice layer interface.

Simulations with a layer with densities of 500 kg/m$^3$ and 600 kg/m$^3$ (Figure 4d and g, Figure S3) reveal much smaller peak offset (e.g. -2.4 m to 1.7 m for a 0.1 m thick 600 kg/m$^3$ layer) compared to those with a typical ice layer density of 862 kg/m$^3$ (-6.5 m to 8.3 m). This suggests that the highest $dz$ values only occur when ice layers are present in the firn.





Figure 4: a) Surface peak offset (dz) for RadSPy simulated MCoRDS surface returns over firn stratigraphies consisting of a single ice layer placed at various depths and with different thicknesses. b) and c) show example model input profiles. d) dz for a 0.1 m thick ice layer, as well as layers with densities of 500 kg/m³ and 600 kg/m³ at different depths, with waveforms for selected ice layer depths (marked with crosses) shown in e and f. g) dz for an ice layer, as well as layers with densities of 500 kg/m³ and 600 kg/m³, starting at 1 m depth and with increasing layer thickness, with waveforms for selected ice layer thicknesses (marked with crosses) show in h and i. The dotted lines in the waveform plots show the picked peak identified used to calculate dz.

Figure 5 shows the modelling results over a firn structure with multiple, evenly spaced ice layers of varying thicknesses. Here, dz tends to be highest when firn contains 3-5 ice layers (i.e. 6-10 strong density contrasts), however, dz is also dependent on the thickness of the ice layers (Figure 5a).

Our modelling experiments show that over vertically homogeneous stratigraphies, such as in the ablation and dry-snow zones, the surface reflection peak accurately returns the true ice sheet surface ($dz \approx 0$). Conversely, over vertically heterogeneous stratigraphies, the surface reflection peak often deviates from the true ice sheet surface. While positive dz





values are more common, specific combinations of ice layer depths and thicknesses can result in negative *dz* values. Overall,

these results confirm the use of non-zero peak-offsets (*dz* ≠ 0) for identifying areas with a vertically heterogeneous firn stratigraphy within top ~8 m. However, it is important to note that the same *dz* value can be produced by multiple heterogeneous firn stratigraphies, limiting the ability to infer specific firn characteristics based solely on *dz*.

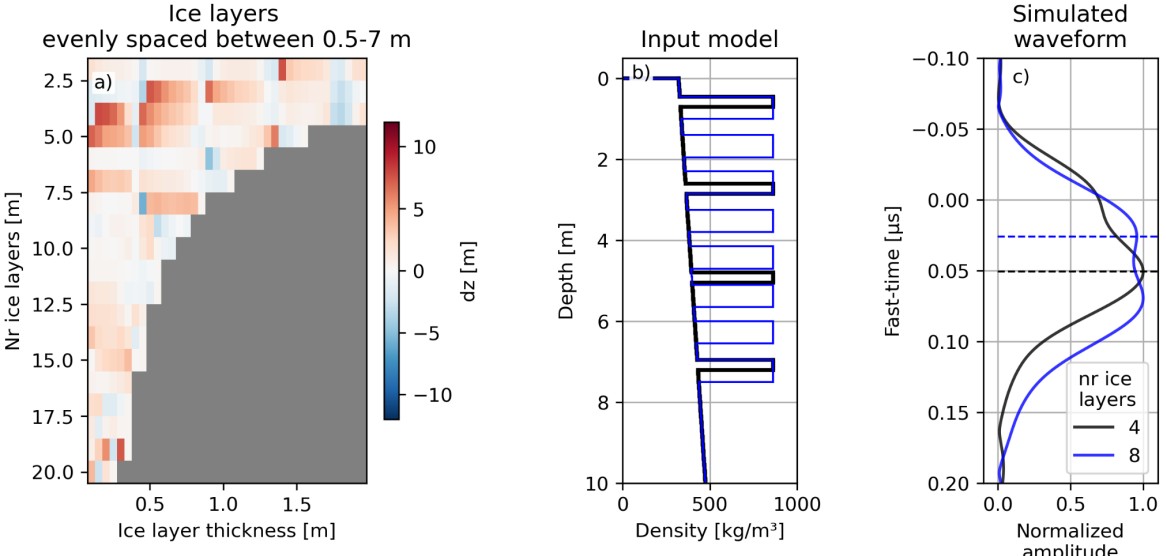

**Figure 5: a) Surface peak offset (dz) for RadSPy simulated MCoRDS surface returns over firn stratigraphies with multiple ice layers of varying thickness. b) Two example input models, and c) the resulting simulated waveforms.**

## 3.1 Effects of waveform sampling

The shape of the waveform measured by MCoRDS is influenced by the radar's sampling rate. To assess its impact on the identification of the surface reflections and the resulting peak offsets (*dz*), we conducted tests using downsampled RadSPy-

simulated waveforms in two scenarios: i) a waveform with a single peak originating from a homogeneous subsurface (Figure 6a), and ii) a waveform with a double peak, generated by a firn profile containing a single ice layer (Figure 6d). These RadSPy waveforms are downsampled to match the MCoRDS sampling rate (33.3 ns), but with varying time-zero offsets (0-33 ns) to simulate different distances between the radar receiver and the ice sheet surface.

For waveforms with a single peak, downsampling produces only minor variations in the waveform shape. When employing our upsampling approach to identify the surface reflection peak, the location of the identified peak shows minimal fluctuation, with *dz* offsets ranging between 0-0.5 m (Figure 6a-c). In contrast, waveforms with double peaks may not be fully resolved at the MCoRDS sampling rate, making them irrecoverable even with the upsampling approach (Figure 6f). In such instances, the peak offset *dz* fluctuates around ±6 m, where negative values occur when both peaks are captured, and





positive values arise when the first peak is missed during sampling. These examples suggest that any bulges observed at the leading edge of a peak reflection in MCoRDS signal (e.g. Figure 2c) likely represents the air-ice sheet interface. Therefore, using an upsampling approach to pick this reflection is a reasonable strategy.

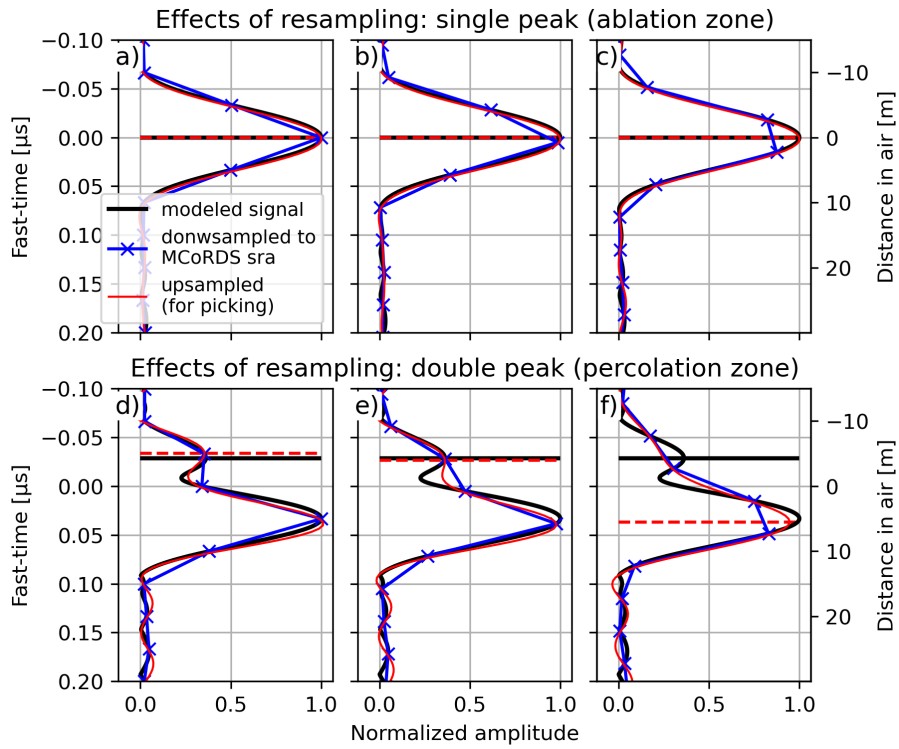

Figure 6:  **Effects of downsampled signals to the MCoRDS sampling rate (sra) with different starting lags (0 ns left panels, 5 ns middle panels, 15 ns right panels). a-c) show the RadSPy modelled waveform over the ablation zone (air-ice), and d-f) show the modelled signal over the percolation zone including a 10 cm thick ice layer at 2.1 m depth. Horizontal lines show which peak was identified from the modelled signal at original sampling rate (black) and downsampled signals (red dashed).**

## 4    Observed radar surface peak offsets

### 4.1    Spatial pattern of surface peak offsets and surface reflection characteristics

Figure 7 presents yearly gridded radar surface peak offsets (*dz*) over the GrIS between 2011-2019 (see Figure S4 for non-gridded *dz* along profile lines). While there are some annual variations (discussed in Section 4.3), the general spatial pattern of *dz* is consistent over all survey years. Specifically, low *dz* values (<1 m) are observed near the ice sheet margin and over the high-elevation interior, while a band of increased *dz* (>1 m) is evident at mid-range elevations. The transition from low to

high *dz* is relatively abrupt at lower elevations, aligning closely with the onset of the percolation facies, which here is derived where the mean 2009-2019 Modèle Atmosphérique Régional (MAR, (Fettweis et al., 2017)) surface mass balance (SMB)



equals zero (dashed lines in Figure 7). In the northern and northeastern regions, this transition lies at elevations slightly below the percolation facies. The transition from high to low *dz* at the upper percolation zone is more gradual and spreads across the boundary to the dry-snow facies. The dry-snow facies is derived where the mean 2009-2019 MAR melt equals 50 mm w.e./yr, outlining areas that receive little melting. Using all data from 2011-2019, we find that the median *dz* values (± its standard deviation) in the ablation, percolation- and dry-snow zones are 0.1±1.9 m, 2.2 ± 2.7 m, and 0.3 ± 1.2 m, respectively (Figure S5).



**Figure 7: Maps showing the radar surface peak offsets (*dz*) over the GrIS between 2011-2019. For improved visualisation, the data has been gridded onto a 10x10 km grid, with data interpolated over empty grid cells less than 30 km from the nearest grid point with data (see Figure S4 for non-gridded *dz* values along the radar profiles). Dark orange contour line represents *dz* of 2 m. The dashed black line indicates the boundary between the ablation- and accumulation zones (derived from MAR SMB), and the dotted black line depicts the dry-snow facies (derived from MAR melt). Thin black lines are elevation contour lines at a 500 m interval.**



Figure 8 shows *dz* along an MCoRDS profile (Profile A) collected in western Greenland in 2017 (see Figure 7e for profile

location), in conjunction with Accumulation Radar (AR) data. Lower *dz* values over the ablation- and dry-snow zones

correspond to clear, narrow surface return peaks in the MCoRDS signal (Figure 8d). Notably, the increase in *dz* at the

transition between the ablation- and percolation zones coincides with the upper elevation range of previously mapped ice

slabs (Jullien et al., 2023; MacFerrin et al., 2019) (Figure 8a, Figure S4). A subsequent comparison reveals that *dz* decreases

as ice slab thickness increases, stabilising at zero for ice slabs thicker than ~8 m (Figure S6).

The AR radargram in the upper dry-snow zone displays numerous bright, continuous reflectors, likely representing

isochronous layers of past summer surfaces (Lewis et al., 2017). In contrast, the lower percolation facies features more

diffuse reflectors (Figure 8b), likely a result of annual surface melting causing the coalescence of individual ice layers, ice

pipes between ice layers (Humphrey et al., 2012) and an overall higher firn ice content hampering the radar detection of

annual layers (Culberg et al., 2021). These characteristics are mirrored in the MCoRDS surface signal (Figure 8c), where the

surface reflection broadens (in fast-time) and often splits into multiple peaks (Figure 8d), resulting in high and spatially

variable *dz*. Figures S7 and S8 present additional profiles (B, C) comparing *dz* to MCoRDS and AR radargrams.



**Figure 8: Radar surface peak offset (*dz*) over OIB CReSIS MCoRDS and Accumulation Radar (AR) profile A in Central West**
**Greenland (see Figure 7e for profile location). a) Ice sheet surface elevation (black) and *dz* (raw in light blue, and smoothed over a**
**1 km window in dark blue) along the profile. Vertical dotted black/white lines indicate the firn zone transitions (ablation zone,**
**percolation zone, dry-snow zone) derived from MAR, and the blue shaded area marks the location of previously mapped ice slabs**
**(Jullien et al., 2023). b) AR data showing the firn stratigraphy, including ice slabs and isochronous layers of the uppermost 20 m.**
**c) MCoRDS data of the uppermost 50 m in firn/ice. The radargrams have been flattened with respect to the picked surface**
**reflection. d) MCoRDS traces along the profile showing the shape of the surface reflection along the profile. The traces are aligned**
**with respect to the picked surface reflection, and normalised to the maximum energy.**

In the dry-snow zone, a few localised areas exhibit elevated *dz* values that persist over multiple survey years. For instance,

increased *dz* values near the summit region are consistent from 2012-2019 (Figure 7). However, AR radargrams over these

locations do not indicate abrupt changes in the near-surface stratigraphy (Figure 9, also profile E in Figure S9). Rather, the

localised *dz* increase seems to correspond to a region where the MCoRDS surface signal consists of a broadened single peak,

laterally bounded by areas with separated double-peaks.





**Figure 9: Radar surface peak offset (*dz*) over OIB CReSIS MCoRDS and Accumulation Radar (AR) profile D over the *dz* anomaly region near the summit (see Figure 7b for profile location). a) Ice sheet surface elevation (black) and *dz* (blue) along the profile. b) AR data showing the firn stratigraphy of the uppermost 20 m. c) MCoRDS data of the uppermost 50 m in firn/ice. The radargrams have been flattened with respect to the picked surface reflection. d) MCoRDS traces along the profile showing the shape of the surface reflection along the profile. The traces are aligned with respect to the picked surface reflection, and normalised to the maximum energy.**

## 4.2 Correlation with in-situ firn properties

To investigate the relationship between radar surface peak offsets (*dz*) and in-situ firn conditions, we use four variables derived from the 28 spatially and contemporaneous firn cores (Section 2.5). These include the mean density, the standard deviation of the density, the number of density contrasts greater than 50 kg/m$^3$, and the total amount of ice content, defined as the length percentage of core sections with density exceeding 862 kg/m$^3$ (Rennermalm et al., 2021). The core stratigraphies used for the comparison are shown in Figure S10.

The correlation between *dz* and the mean core density exhibits two opposing trends (Figure 10a), complicating any direct inversion of *dz* to firn density. However, *dz* shows a linear increase with both the standard deviation of the density profile





(Figure 10b) and the number of density contrasts (Figure 10c). We tested different density contrasts thresholds ranging from 5-200 kg/m³ and found that the linear relationship starts to emerge for contrasts above ~50 kg/m³ (Figure S11). These

parameters can be viewed as proxies for the vertical heterogeneity of the firn, and thus, support the hypothesis that high $dz$ values are indicative of vertically heterogeneous firn profiles.

No clear correlation is found between $dz$ and the firn ice content (Figure 10d). While firn cores lacking any ice (all recorded densities < 862 kg/m³) generally show $dz$ <1.3 m, high ice content levels (>10 %) typically result in $dz$ values ranging from ~1.3-4 m. The sole exception is the 2017 summit firn core which yielded a $dz$ of 2.2 m for an ice content of 0 % (Figure

S10).

Overall, low $dz$ values appear to correspond with relatively uniform vertical firn (or ice) stratigraphy, while high $dz$ values are associated with vertically heterogeneous firn profiles. However, the non-uniqueness of $dz$ in relation to firn density and ice content makes it challenging to quantitatively relate $dz$ to specific firn properties.

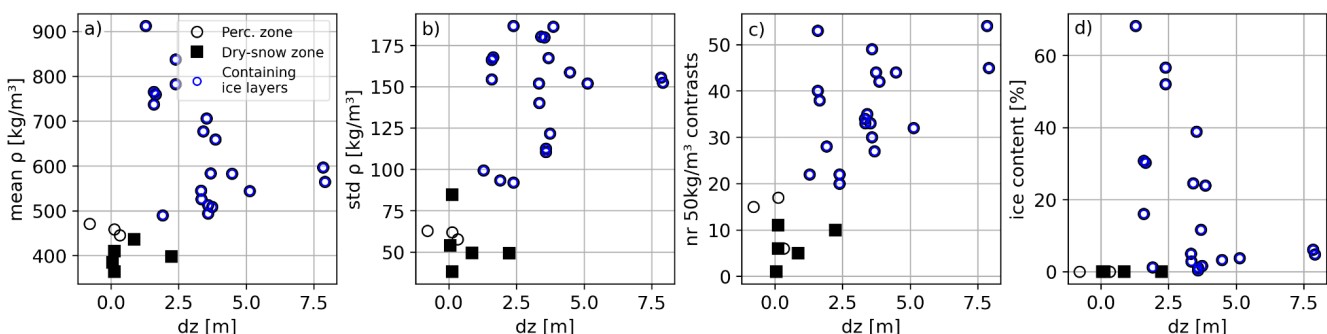


**Figure 10: Radar surface peak offset ($dz$) versus firn core properties (extending from surface to 7.5 m depth): a) mean density, b) the standard deviation of the firn density, c) the number of density contrasts above 50 kg/m³, and d) the firn ice content (calculated as length percentage of firn density exceeding 862 kg/m³. Blue markers indicate firn cores that contain ice layers.**

### 4.3 Response of radar peak offset to inter-annual climate variability

To analyse temporal variations in the near-surface facies, we establish an empirical $dz$ boundary of 1 m based on our modelling tests and firn core comparisons. Specifically, $dz$ <1 m indicates a vertically homogeneous structure, such as massive ice slabs or firn in the dry-snow zone, while $dz$ >1 m suggests a vertically heterogeneous structure, typically firn with single or multiple ice layers in the percolation zone. The outcomes of this classification and the most significant firn changes are presented in Figure 11.

Between 2011-2013, areas of heterogeneous firn ($dz$ >1 m) expanded upwards in elevations across all sectors of the GrIS, reaching their maximum extent in 2013 (Figure 11 and Figure 12). Particularly over the period from 2011-2012, $dz$ values increased in the percolation zone in the CW region (Figure 12 and S12). We note that in the southern region, the year of maximum heterogeneous extent is less certain due to limited data coverage in 2012/13, but was no later than in 2014. During the same period (2011-2013), the lower boundary separating homogeneous and heterogeneous facies near the



ablation/accumulation zone transition also shifted upwards, especially in the northwestern regions (~150 m in CW and ~300 m in NW and NO, Figure 12). Furthermore, we note that in the year 2013, the spatial extent of the *dz*-derived heterogeneous firn area matches well with the locations where the 2012 melt layer has been mapped in the firn (using a layer prominence > 0.5 (Culberg et al., 2021)) (Figure 11c).

Comparing the minimal extent of *dz*>1 m in 2011 to its maximum extent over the period 2012-2014, a total area of 338,450

km$^2$ located in the upper percolation- and lower dry-snow zones switched from homogeneous firn to heterogeneous firn after spring 2011 (Figure 11g). This expansion of heterogeneous firn follows years where the ice sheet experienced relatively high surface melting, with the extreme melt event in summer 2012 (Figure 12).

Between 2013 and 2014, areas of heterogeneous firn in the dry-snow zone reverted back to homogeneous firn (Figure 12d), aligning with a year of reduced melting in 2013. This trend continued through 2019, with additional areas in the upper

percolation zone reverting to homogeneous firn. Absolute *dz* values generally decreased after 2013, but saw an intermittent increase in the southwestern regions from 2014 to 2017, corresponding to relatively high melt years in 2015 and 2016, followed by another decrease from 2017-2019 (Figure S12).

Comparing the maximum extent of *dz* >1 m during 2012-2014 to its minimum extent during 2017-2019, a total area of 664,734 km$^2$ located in the upper percolation- and dry-snow zones transitioned from heterogeneous to homogeneous (Figure

11h). This change was most pronounced in the southern region and followed years of generally reduced surface melting (Figure 12).

Repeat firn cores collected from 2015-2017 at NASA South-East (NSE) and Saddle (SDL, locations in Figure 11e) offer insights into in-situ firn changes where *dz*-classified firn transitioned from heterogeneous to homogeneous between 2014-2019. Both locations exhibited a decline in the standard deviation of firn density between 2016-2017, which for SDL already

started after 2015, indicating a shift towards more homogeneous vertical firn profiles (Figures S13 and S14).

Lastly, while heterogeneous firn continued to retreat in the southwestern regions between 2017-2019, it expanded in the northeastern (NE) sector (Figure 11e and f), coinciding with a slight positive melt anomaly in 2017 (Figure 12). This expansion of heterogeneous firn in NE Greenland aligns with reported increase in near-surface densities in 2018 (Scanlan et al., 2023).






**Figure 11: a-f) Qualitative classification of radar surface peak offset (*dz*) into homogeneous firn/ice (*dz*<1m, blue) and heterogeneous firn (*dz* >1m, orange) areas. The dark blue and red shaded areas represent the change in classification from heterogeneous to homogeneous, and vice versa when compared to the previous survey year. The yellow dots on c) show locations where the ice layer prominence is above 0.5, indicating the ice layer formed by the 2012 extreme melt likely occurs in the firn (Culberg et al., 2021). The triangles on e) show the firn core locations at NASA South-East (NSE) and Saddle (SDL). g) Change in *dz* classification between 2011-2012/13/14, highlighting the switch from homogeneous to heterogeneous firn in the upper percolation zone/dry-snow zone (red). The years 2012-14 are combined to represent the maximum extent of heterogeneous firn. h) Change in *dz* classification between 2012/13/14 and 2017/19, highlighting the switch from heterogeneous to homogeneous firn in the upper percolation zone/dry-snow zone (blue). The 2017/19 data is combined to represent the minimum extent of heterogeneous firn.**



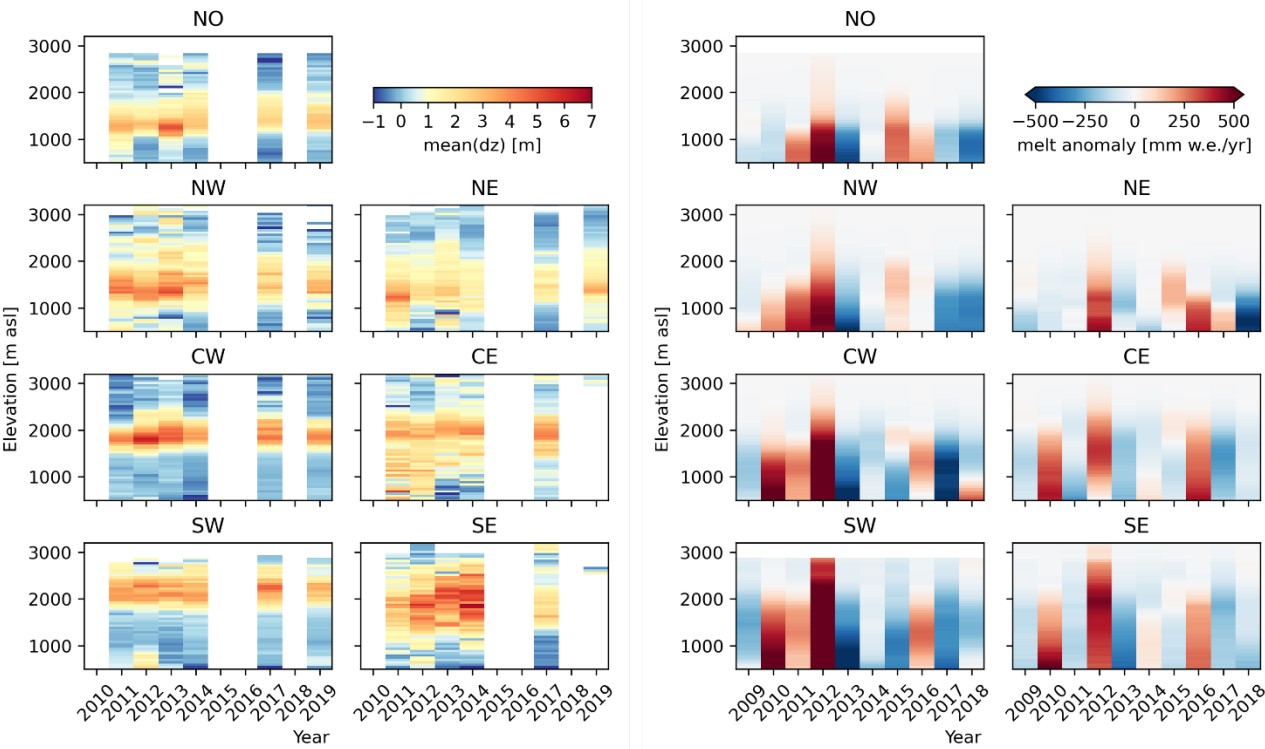

**Figure 12: Left: Spatio-temporal change in radar surface peak offset (*dz*) between 2011-2019. *dz* values are binned into 50 m elevation bands separated by region, where the colour represents the mean *dz* value in the elevation bin measured in a specific year. Right: MAR-derived yearly melt anomalies from the mean annual melt rate between 2009-2018, binned into elevations same as on the left.**

## 5 Discussion

Through the combination of modelling and observations, we have shown that it is possible to utilize the spatial distribution of surface reflection peak offsets (*dz*) to effectively delineate between vertically homogeneous and heterogeneous firn profiles across the GrIS. Our approach provides a rapid and consistent method for identifying firn influenced by surface melting and refreezing processes.

The results reveal an upward expansion of heterogeneous firn into the percolation- and dry-snow zones between 2011-2013, followed by a reduction of heterogeneous firn between 2013-2019. The expansion of heterogeneous firn in the percolation and dry-snow zones between 2011-2013 is attributed to the formation of new ice layers within formerly homogeneous firn, following surface melt and refreeze events during the summers of 2011 and 2012. The good spatial alignment between the 2012 melt layer, which was typically found at 1 m depth in 2013 (Culberg et al., 2021), and the heterogeneous firn region identified in 2013 further substantiates that *dz* is sensitive to the presence of ice layers.



The switch of heterogeneous firn back to homogeneous firn over the period 2014-2019 aligns with localised firn pore space replenishment observations from firn core measurements (Rennermalm et al., 2021). These firn cores, collected in the percolation zone of CW Greenland, revealed a decline in the number of ice layers in newly formed firn in the top 3.75 m post-2013 (Rennermalm et al., 2021). Similarly, our analysis of the SDL and NSE firn cores indicated a decline in firn
profile heterogeneity from 2015-2017. Thus, our findings indicate a widespread manifestation of these replenishment processes, covering at least ~664,734 km$^2$. As outlined in prior studies (Rennermalm et al., 2021), this likely stems from low surface melting and high accumulation rates between 2017-2018, culminating in the most positive mass balance anomaly compared to the 2003-2013 period (Sasgen et al., 2020). Furthermore, under such high mass balance conditions, the 2012 melt layer became gradually buried, reaching depths between 3-12 m by 2017 (Culberg et al., 2021), and surpassing 7.5 m in
CW Greenland by 2018 (Rennermalm et al., 2021). Collectively, the observed reduction in *dz* and inferred switch from heterogeneous to homogeneous firn can most likely be attributed to the burial of the 2012 melt layer beyond the MCoRDS range resolution, and the overall decrease in melt layers and density contrasts within the firn.

The lower boundary between homogeneous and heterogeneous firn generally aligns with ice slabs around 6-7 m thick. The
upward shift of this boundary from 2011 to 2019 implies that thin ice layers in the firn have grown into ice slabs as well as overall thickening of ice slabs as documented by Jullien et al. (2023). As supported by the modelling results, when these ice slabs grow to occupy the top ~8 m in firn, the radar signal perceives the surface as homogeneous, thereby shifting this boundary towards higher elevations as ice slabs grow with subsequent melt years.

Our findings also have implications for estimates of future ice slab formation and meltwater runoff. The presence of thin ice layers in firn can facilitate the formation of thick, impermeable ice slabs in subsequent melt years (Jullien et al., 2023; MacFerrin et al., 2019). Our results indicate that the 6 year period (2013-2019) with lower surface melting allowed firn in the high percolation zone to recover after the 2012 extreme melt event. If the 2012 extreme melt year was followed by another extreme melt event before the heterogeneous firn could be replenished, the consequent formation of ice slabs would
likely have modified meltwater volumes. Thus, outlining regions with pronounced peak offsets (*dz*) indicating the presence of ice layers in firn can help identify areas predisposed to future ice slab formation (e.g. dark orange contours for *dz* > 2 m on Figure 7). Additionally, identifying firn with ice layers is essential for interpreting satellite radar measurements, as internal reflectors can generate ambiguous signals (Nilsson et al., 2015).

Our results also reveal areas with complex surface processes, for example, localised high *dz* areas in the dry-snow zone, particularly near the summit of the ice sheet. Here, slightly higher firn density compared to neighbouring regions could be inferred from the trend of increasing *dz* with increasing bulk firn density up to 600 kg/m$^3$ (Figure 10). This aligns well with the unexpectedly high surface densities reported by (Scanlan et al., 2023). The persistence of this anomalous feature over



several years is unclear, but may be attributed to the relatively low snowfall in the area, implying that the near-surface varies
only over long timescales.

While our modelling and observations generally align, some discrepancies exist. For instance, modelling predicts $dz = 0$ over
the ablation- and dry-snow zones, yet our observations also include non-zero (and negative) values in these areas. Such
deviations could stem from factors not accounted for in the model, such as the surface slope (given that the laser is nadir
looking, versus the radar records the nearest return, which may result in negative $dz$ values over sloping surfaces) surface
roughness and surface anomalies like crevasses. Further, the non-unique nature of $dz$ across different firn stratigraphies,
especially in the percolation zone, limits the extraction of quantitative firn properties such as the density or ice content from
$dz$.

Altogether, our results demonstrate that airborne VHF radar measurements, when combined with laser altimetry, can
effectively discern different firn facies. This provides opportunities to assess firn properties in situations where only VHF
radar is accessible (i.e. in the absence of UHF radar or in-situ observations), particularly in contexts predating Operation Ice
Bridge's Accumulation Radar. Furthermore, analogous techniques may prove invaluable for upcoming Earth-orbiting radar
sounders when combined with existing laser altimetry missions (e.g. ICESat-2), as well as interplanetary missions such as
ESA's JUpiter ICy moons Explorer (JUICE) spacecraft (Grasset et al., 2013), which carries both a radar sounder (RIME)
and a laser altimeter (GALA). Future work could extend this methodology to higher frequency ranges, as well as leverage
satellite radar and laser altimetry for snowpack property delineation. Finally, we suggest that similar and potentially further
firn properties could be derived by examining other surface reflection characteristics, such as the width and peakiness similar
to applications for characterising subglacial environments (e.g. Oswald and Gogineni, 2008; Jordan et al., 2017).

## 6   Conclusions

In this study, we combined airborne radar sounding and laser altimetry measurements to derive radar surface reflection peak
offsets ($dz$) over the GrIS. Our results, supported by modelling and in-situ firn core analyses, demonstrate that $dz$ serves as
an effective tool to delineate between vertically homogeneous and heterogeneous firn profiles, where temporal changes in $dz$
align well with known climatic events. This allows for a nuanced understanding of spatial and temporal variations in firn
stratigraphy from 2011-2019, highlighting firn regions with important implications for meltwater runoff and ice sheet mass
balance. For the first time, our results map and quantify areas (664,734 km$^2$) where firn replenishment took place between
2014-2019, previously observed only in localised in-situ measurements, emphasising the importance of spatially
comprehensive observations. Importantly, $dz$ can be used to outline firn areas that are predisposed to future ice slab
formation, which has considerable impact on the firn's future meltwater storage capacity.



## Data availability

The MCoRDS rds and accumulation radar data used in this study are available on the CReSIS public webpage https://data.cresis.ku.edu/. The IceBridge ATM data is available through NSIDC at https://nsidc.org/data/ilatm2. SUMup firn density measurements (Thompson-Munson et al., 2022) were accessed from the Arctic Data Center (https://arcticdata.io/). Ice slab data (Jullien et al., 2023) were derived from https://zenodo.org/record/7505426, firn aquifer

data from the Arctic Data Center (https://arcticdata.io/catalog/view/doi:10.18739/A2TM72225), and the snowline dataset (Fausto et al., 2018) from https://thredds.geus.dk/thredds/catalog/MODIS_snowline/catalog.html. MAR data (Fettweis et al., 2017) was accessed from ftp://ftp.climato.be/fettweis. GrIS basins (Mouginot and Rignot, 2019) were derived from https://doi.org/10.7280/D1WT11. All data products resulting from this work are currently will be available on Dataverse (https://dataverse.geus.dk/) upon publication.

## Author contributions

A.R. conceptualized the study with inputs from all co-authors and led the methodology development and research investigations. K.M.S. setup the modelling experiments, B.V. assisted with firn core analyses and P.H. assisted with software implementations and data curation. N.J. provided ice slab data resources and insights for interpreting the data. A.R. drafted the original manuscript and visualizations. All co-authors provided critical reviews and contributed to the final manuscript.

A.P.A acquired the funding supporting the study.

## Competing interests

At least one of the (co-)authors is a member of the editorial board of The Cryosphere.

## Acknowledgements

We acknowledge the use of data from CReSIS generated with support from the University of Kansas, NASA Operation

IceBridge grant NNX16AH54G, NSF grants ACI-1443054, OPP-1739003, and IIS-1838230, Lilly Endowment Incorporated, and Indiana METACyt Initiative. We employed OpenAI's GPT-4 language model (ChatGPT-4) for text editing.



## Financial support

This study and all personnel at the Geological Survey of Denmark and Greenland was supported by the Greenland Climate
Network (GC-Net) monitoring program. K.M.S. was supported by through Villum Fonden (Villum Experiment Programme)
Project No. 50225, and N.J. was supported by European Research Council award 818994 (CASSANDRA).

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
