# Peer review of "Mapping the vertical heterogeneity of Greenland's firn from 2011-2019 using airborne radar and laser altimetry"

_EGUsphere, 2023_

## Referee Comment (RC2)

685

[referee-annotated manuscript omitted]

---

## Author Comment (AC1)

**Review from Anonymous Referee #1**

This manuscript describes a new method for characterizing the vertical heterogeneity of Greenland's firn using VHF radar sounding and coincident laser altimetry measurements. Using full-wave electromagnetic modeling, the authors show that the offset in the apparent surface elevation measured by these two instruments increases significantly when one or more ice layers are present within the range resolution of the radar system. They demonstrate that this is due to constructive and destructive interference amongst layers that shifts the apparent location of the surface echo in the radar sounding data. The authors then map this offset across Greenland using OIB MCoRDS and ATM data collected between 2011-2019 and discuss the spatial and temporal trends in the context of regional climate. In particular, they show that their method captures both the inland expansion of ice layers following extreme melt in 2012 and the later replenishment of pore space in Southwest Greenland during a period of cooler summers from 2014-2018.

This is a technically complete and well-written paper that introduces an interesting new method for studying the heterogeneity of Greenland's firn. It is exciting to see that that the apparent reduction in shallow ice content since 2012 holds at a larger scale than the initial firn core studies in Southwest Greenland, and this seems like a promising method for tracking trends in percolation zone extent and structure. I have some comments on framing and small details, but overall, I think this is an excellent piece of work.

Thank you much for your time and effort reviewing our manuscript! We truly appreciate the detailed and constructive suggestions, and we agree with all of them and have implemented them in the manuscript. We believe that these changes further improved the manuscript.

Please find answers to all comments below, using the following color code:

Blue indicates comments from the authors

*Green italic indicates changes in the manuscript, with revised text indicated marked as underscored*

**Major Comments:**

**[1]** The motivation for the study, and particularly why the use of VHF radar sounding together with altimetry, did not come across clearly to me in the introduction. The authors list a slew of methods that are currently applied to study firn properties, but do not clearly establish what gap their method is then trying to fill. I think this is particularly important in Greenland because UHF Accumulation Radar data that can directly resolve the firn structure is available in all but one of the years that this study presents, so it is not immediately clear why one would want to use the VHF and laser altimeter combination instead. I think there may be some compelling reasons – for example, portability to Antarctica, where almost no Accumulation Radar data was collected – but the paper would be strengthened if the authors laid those motivations out in the introduction.

Thank you for this comment, we agree that the motivation of using VHF radar was not stated clearly. We added the following to the introduction section:

*"While extensive UHF radar data is available over Greenland and could technically be used to resolve the firn stratigraphy, our proposed VHF radar based method is relatively simple and allows for a classification of firn zones without tracing internal layers or structures from the radargram, and is transferrable to areas such as Antarctica or other icy planetary surfaces, where no or little UHF radar data is available."*

**[2]** Might surface roughness also modulate dz? I could image that for the radar, coherent integration of reflections from facets with different heights could set up constructive or destructive interference patterns (similar to layers at different depths) that might shift the surface height from the expected average of the facet heights. The anomalously high dz values in a few spots in the interior make me wonder if features like sastrugi might impact the retrievals at all.

This is a great comment and made us realize that we only touched on the effects of surface roughness briefly. We agree that this could be a possibility causing the anomalous dz values in the dry-snow zone. Together with comments from below as well as the other reviewer, we changed the statement in the discussion section to the following:

*"Our results also reveal areas with complex surface processes, for example, localised high dz areas in the dry-snow zone, particularly near the summit of the ice sheet. It is possible that these areas consist of a high small-scale density variability in the near-surface (e.g. from wind scour or hoar formation) (Hörhold et al., 2011), causing strong enough density contrasts to affect the radar surface peak. Additionally, increased dz may be caused by increased surface roughness (e.g. from sastrugi), which introduces waveform interferences when the radar signal from facets with different heights when the signal is coherently integrated during processing. We note that this observation aligns with unexpectedly high surface densities in the summit region reported by Scanlan et al., (2023). The persistence of this anomalous feature over several years is unclear, but may be attributed to the relatively low snowfall in the area, implying that the near-surface varies only over long timescales."*

**[3]** What is the typical spread of values within the each 10km grid cell in Figure 7? Does the variability in dz provide any additional information, particularly in terms of the quality of the retrieval? To me, it seems like a potentially useful quantification of uncertainty in the boundaries between radar facies, which is somewhat lacking in the current version of the paper. I could also imagine that dz would be highly variable in the percolation zone, but maybe the dz anomalies in the interior would show a more consistent bias if they're formed by some localized by consistent density anomaly.

Thank you for this great comment and idea for analysis. As suggested, we performed an analysis of the standard deviation of *dz* over each grid cell (see figure below).

[Figure]

*Figure SX: Maps showing the standard deviation of the radar surface peak offsets (dz) within each grid cell over the GrIS between 2011-2019. For improved visualization, the data has been interpolated over empty grid cells less than 30 km from the nearest grid point with data. The dashed black line indicates the boundary between the ablation- and accumulation zones (derived from MAR SMB), and the dotted black line depicts the dry-snow facies (derived from MAR melt). Thin grey lines are elevation contour lines at a 500 m interval.*

As expected by the reviewer, there is an increased standard variation in grid cells where *dz* is high, i.e. over the percolation zone. The higher variability of *dz* over the percolation zone also matches the modeling results, which show that only a slight change in layer depth/thickness can significantly affect *dz*. Assuming the layers are not perfectly flat and continuous, *dz* is therefore expected to change a lot over a short distance in the percolation zone. In contrary, the low standard deviation over most of the ablation and dry-snow zones support the fact that *dz* is

not expected to change over a homogeneous subsurface. This can also be observed in Figure 8a, where *dz* along a profile line has high variability over the percolation zone, and lower variability over the ablation and dry-snow zones.

We included the figure in the supplementary material of the manuscript, and added the following text to Section 4.1 in the manuscript:

*"The AR radargram in the upper dry-snow zone displays numerous bright, continuous reflectors, likely representing isochronous layers of past summer surfaces (Lewis et al., 2017). In contrast, the lower percolation facies features more diffuse reflectors (**Error! Reference source not found.**b), likely a result of annual surface melting causing the coalescence of individual ice layers, ice pipes between ice layers (Humphrey et al., 2012) and an overall higher firn ice content hampering the radar detection of annual layers (Culberg et al., 2021). These characteristics are mirrored in the MCoRDS surface signal (**Error! Reference source not found.**c), where the surface reflection broadens (in fast-time) and often splits into multiple peaks (**Error! Reference source not found.**d), resulting in high and spatially variable dz. Figures S7 and S8 present additional profiles (B, C) comparing dz to MCoRDS and AR radargrams. The increased variability of dz can be observed over the entire percolation zone of the GrIS (Figure S9). Combined with the modeling results suggesting that small changes in ice layer thickness or depth can cause large changes in dz, this indicates that the vertical structure of firn over the percolation zone has a high spatial variability."*

**Minor Comments:**

**Line 79:** Note that the MCoRDS documentation gives an empirical windowing factor of 1.53 for a 20% Tukey window on transmit/receive plus a frequency domain hanning window (https://data.cresis.ku.edu/data/rds/rds_readme.pdf). That seems to be more consistent with the current processing parameters for the MCoRDS data – Rodriguez-Morales et al. (2014) reports a processing chain that using a much more aggressive Blackman window on receive, which does not match up with what is in the current processing spreadsheets.

Thank you for catching this. We changed the windowing factor to 1.53, and updated this sentence:

*"For MCoRDS ($k_t = 1.53$, https://data.cresis.ku.edu/data/rds/rds_readme.pdf), the theoretical vertical range resolution is 4.3 m in ice ($\varepsilon_r = 3.15$) and 5.7 m in firn ($\varepsilon_r = 1.8$, corresponding to a firn density of 384.4 kg/m³)."*

Because with this windowing factor the theoretical vertical range resolution is reduced. We updated the range resolution throughout the manuscript. This also affected our comparison to firn cores – before we compared *dz* to the firn stratigraphy down to 7.5 m depth, now we updated this comparison to only 5.7 m depth. The general results did not change, but it allowed us to include one more firn core for the comparison which was shorter than 7.5 m. We updated all Figures and text regarding the firn cores in the manuscript and supplementary material.

[Figure]

*Figure 1: Radar surface peak offset (dz) versus firn core properties (extending from surface to 5.7 m depth): a) mean density, b) the standard deviation of the firn density, c) the number of density contrasts above 50 kg/m³, and d) the firn ice content (calculated as length percentage of firn density exceeding 862 kg/m³. Blue markers indicate firn cores that contain ice layers.*

We used the updated k value to re-compute the theoretical footprint before focusing and updated Table S1 in the supplementary material.

Lastly, this, together with comments from the other reviewer made us realize that the modeled signal has a larger theoretical vertical range resolution. We calculated the theoretical range resolution in firn for the modeled signal to ~11 m (calculated using the peak withd at a 3dB drop from the maximum peak, similar to as the MCoRDS windowing factor for the range resolution is calculated). We attribute the difference between the modeled and observed signal range resolution to the use of different windowing parameters applied in the signal processing, and potentially also due to slightly different input signals. To explain this, we added the following text to the manuscript:

"*We note that the vertical range resolution of the modelled signal, expressed as the pulse width 3 dB down from the peak (similar to the calculation of the MCoRDS windowing factor, https://data.cresis.ku.edu/data/rds/rds_readme.pdf), is ~8.4 m in ice and ~11 m in firn, The larger range resolution is likely due different windowing factors applied in the pulse compression (here we use the standard RadSPy Hanning window), and the input pulse not being an exact replica of the MCoRDS pulse. However, we expect the general behaviour of the modelled signal to be representative of the MCoRDS signal, where depths of layer interfaces should be considered in a relative sense to the different theoretical range resolutions.*"

With this knowledge, we updated Figure 4 (see below) to represent the depth and thickness of ice layers in firn also as a function of the range resolution. From this, it becomes evident that ice layers affect the surface return when in the top 4 m for the observed MCoRDS signal (0.7xRange Resolution), and not the previously stated 8 m. (A similar secondary x-axis was applied to Figure 3, effects of snow depth, to express the thickness as function of range resolution).

[Figure]

Figure 2: a) Surface peak offset (dz) for RadSPy simulated MCoRDS surface returns over firn stratigraphies consisting of a single ice layer placed at various depths and with different thicknesses. b) and c) show example model input profiles. d) dz for a 0.1 m thick ice layer, as well as layers with densities of 500 kg/m³ and 600 kg/m³ at different depths, with waveforms for selected ice layer depths (marked with crosses) shown in e and f. g) dz for an ice layer, as well as layers with densities of 500 kg/m³ and 600 kg/m³, starting at 1 m depth and with increasing layer thickness, with waveforms for selected ice layer thicknesses (marked with crosses) show in h and i. The dotted lines in the waveform plots show the picked peak identified used to calculate dz. *The x-axis on the top of panels d) and g) show the ice layer depth and thickness scaled by the rang resolution $\delta_r$ of the modelled signal in firn (~11 m).*

We changed the text in the results section of the manuscript accordingly:

*"When an ice layer lies at depths greater than ~$0.7\delta_r$, thus ~4 m for the MCoRDS signal in firn, the radar's surface return displays two distinct peaks…. Thus, ice layers located deeper than ~$0.7\delta_r$ have negligible impact on the surface reflection peaks, leading to dz ≈ 0."*

**Lines 81 – 82:** this seems not quite right for the various horizontal resolutions. The formula given is for the pulse-limited cross-track resolution, but needs to be noted as such, because the MCoRDS data have undergone synthetic aperture focusing and have a much smaller footprint in the along-track direction. The nominal along-track resolution should be ~25 m after focusing and incoherent summation, and typically the trace spacing is about half that (~14 m as correctly noted here). (Again, see the data documentation: https://data.cresis.ku.edu/data/rds/rds_readme.pdf.)

Thank you for catching this. We changed the text accordingly:

*"The MCoRDS radar data have a nominal pulse-limited cross-track resolution (footprint) at the ice sheet surface (Dpl $= 2\sqrt{\frac{ck_t}{B}}R$, where R is the aircraft height above the ice sheet) of ~180 m, which becomes ~25 m after focusing and incoherent summation, and a typical trace spacing of 14-30 m, depending on the survey year (see Table S1).*

**Line 115:** Since you upsample by a factor of 10 before picking the surface, I would think that the vertical picking error should be +/- ~3 ns which would translate to height offsets of 0.32-0.27 m.

Yes, correct. We changed the text to: "*A vertical picking error of 3.3 ns (one fast-time sample on the upsampled signal) would translate to a surface height offset between 0.28 m (ice) to 0.37 m (firn).*"

**Line 117:** Considering my comments on the along-track resolution, perhaps specify here what the total length of your moving average window actually is.

We changed this accordingly to clarify that we use the pulse-limited footprint before focusing was applied: "… apply a moving average window over the mean pulse-limited radar footprint diameter (~180 m before focusing, calculated for each profile) …"

**Figure 2:** it's hard to see the difference between black and blue in the figure, perhaps do something with more contrast like black and red?

Done, we now use black and red.

**Line 134:** give the cutoff number for the maximum deviation from the straight line here for reproducibility.

We realize that this was not very clear, so we changed it to: *"For the 2011 and 2012 datasets lacking roll angle information, we use the deviation in the heading of 10 km long flight line segments as a roll angle proxy and exclude sections where the change in heading exceeds 0.02°."*

**Line 145:** Are there significant differences in the ablation zone dz offset between different flights within the same season? If so, is only using the west coast for calibration adequate? For example, I might expect some flights in North Greenland not to cross this calibration area.

Yes, there are some survey dates with flight paths that did not cross the calibration area. However, the general distribution of *dz* from these flights appears similar to the ones with data over the calibration area, and we therefore believe it is fair to apply the same calibration value to these datasets. We included a figure in the appendix highlighting this.

Generating this figure, we also realized that we accidentally included a second calibration value for the 2014 data instead of removing a dataset. We also corrected the unnecessary removal of some of the 2019 datasets. We corrected Table S1 accordingly and updated the calibrated dz values and all figures. This re-calibration affected our area estimates of homogeneous/heterogeneous firn areas slightly (mainly due to the changes in the 2014 dataset): Homogeneous to heterogeneous firn switch over an area of 350,815km$^2$ (instead of 338,450

km$^2$), and a heterogeneous to homogeneous firn switch over an area of 667,725 km$^2$ (instead of 664,734 km$^2$). We updated these numbers throughout the manuscript.

[Figure]

*Figure S 2: Median peak offset (dz) values for each survey date using all data (black) and only data over the calibration area (red). Manually removed datasets are marked with grey circles.*

**Line 148:** Is the distribution generally close to gaussian in these 10 km grids cells? Is there much difference if you look at the median vs mean?

Thank you for the idea of doing some statistics on the *dz* values within the grid areas. The mean and median are generally similar over the grid cells, indicating mostly gaussian distribution (see Figure below showing the mean minus median in each grid cell). We therefore argue that it is fair to use the mean *dz* to calculate the grid cell values.

[Figure]

**Line 161:** Is a citation to a paper missing here? Courville & Perry (2021) seems to be a software, so it's not clear what "Equation 15" is in that context. The software also appears to use csv input of the ideal transmit waveform, rather than an equation.

Yes, thank you for catching that. We added the correct reference to the publication now:

*Courville, S. W., Perry, M. R., and Putzig, N. E.: Lower Bounds on the Thickness and Dust Content of Layers within the North Polar Layered Deposits of Mars from Radar Forward Modeling, The Planetary Science Journal, 2, 28, https://doi.org/10.3847/psj/abda50, 2021.*

**Line 195:** does variability in the dry firn density matter? In the dry snow zone that can lead to a standard deviation of small scale density variations on the order ~35 kg/m^3 near the surface, which maps to plausible density contrasts of up to ~100 kgm^3 just from hoar formation and wind scour (Hörhold et al., 2011) (see the North Greenland Traverse cores high resolution measurements as an example). At least one previous statistical model of typical firn profiles

did try to take this into account, but was also looking primarily at ice lenses impacts (Culberg & Schroeder, 2020).

Yes, thank you for this great comment. We agree that smaller density contrasts can affect dz. We included this in the discussion section when talking about the *dz* anomalies in the dry-snow zone (also see above):

*It is possible that these areas consist of a high small-scale density variability in the near-surface (e.g. from wind scour or hoar formation) (Hörhold et al., 2011), causing strong enough density contrasts to affect the radar surface peak.*

**Line 246:** Okay, generally answers my question above, but given that later on your median offset in the percolation zone is only 2.2 m, it does make me wonder dry firn variability matters (and might be another explanation for the strange areas of large dz in the dry snow zone).

Addressed in comment above

**Section 3.1:** not suggesting that you should actually do this for this particular paper, but it is possible to reprocess the OIB MCoRDS data with a higher vertical sampling rate. The posted data has been downsampled by at least a factor of 2 in the vertical and is not the native sampling rate of the radar. It might be worth mentioning in the discussion or future work for others considering using this method.

Thank you for pointing out this possibility. We added a sentence to the discussion section mentioning this option for future studies:

*"Further, the OIB MCoRDS data could be reprocessed with a higher vertical sampling rate, which could eliminate the need for our upsampling approach to identify the surface reflection."*

**Figure 9:** can the snow radar tell you anything about what is going on here with its higher resolution? For example, rule in or out very thin layers of higher density near the surface due to wind scour or other such processes?

We appreciate the idea of looking at the snow radar data. However, we leave this for future work, but mention the possibility of small-scale density contrasts in the paragraph on the dry-snow zone anomalies (see above).

**Line 402:** might be an appropriate place to also point to (Rennermalm et al., 2021)?

We don't point to the Rennermalm reference here because these firn cores were not part of that study. We leave it as is and discuss the Rennermalm cores in the discussion section.

**Line 465:** I am not sure that this mapping method adds much to this beyond showing the limits of the percolation zone.

We agree that we mainly use dz to map the percolation zone, but we believe that it is valuable to "spell out" how identifying ice layers in firn can be useful for understanding future firn changes. We therefore keep the sentence in the manuscript.

**Line 473:** Summit and NEEM both had evidence of thin ice crust/layer formation after 2012, so if it is indeed being buried slowly, it could maybe explain some of these anomalies (though doesn't help with what's already there in spring 2012) (Nghiem et al., 2012).

Agreed. To include this possibility, we changed the paragraph about the dry-snow anomalies (see revision from above).

**Line 493:** Given that dz often has something to do with multiple peaks in the radar data, I expect this might work really well for capturing the same kinds of properties and would let you get around the need for coincident laser altimetry collection, which would be great.

Yes, exactly. We hope to test this in future work. We added the following to this sentence:

"*Finally, we suggest that similar and potentially further firn properties could be derived by examining other surface reflection characteristics, such as the width and peakiness similar to applications for characterising subglacial environments (e.g. Oswald and Gogineni, 2008; Jordan et al., 2017), which could also eliminate the need for concurrent laser observations.*"

**References added:**

Hörhold, M. W., Kipfstuhl, S., Wilhelms, F., Freitag, J., and Frenzel, A.: The densification of layered polar firn, Journal of Geophysical Research: Earth Surface, 116, https://doi.org/10.1029/2009JF001630, 2011.

---

## Author Comment (AC2)

**Response to review from Tate Meehan**

Within, "Mapping the vertical heterogeneity of Greenland's firn from 2011-2019 using airborne radar and laser altimetry" the authors devise a radar surface echo retrieval for shallow firn facies classification. The manuscript is well developed and utilizes modeling and in-situ data to improve radar signal interpretation. Much thought is given to the electromagnetic modeling set-up regarding the presence of snow cover on ablated ice and ice layers within the percolation zone of the firn. Modeling results provide a compelling argument for the divergence of radar surface echo waveforms and the misclassification of surface picks based solely on maximum amplitude. When the surface echo retrieval is applied ice sheet wide, interesting and confirmatory information is discovered, such as the replenishment of homogeneous firn within the percolation zone after the significant 2012 melt event. Results from this radar retrieval approximately confirm facies boundaries established by reanalysis model results, but I am left wanting a bit more analysis on how closely the radar and reanalysis resemble each other in a spatiotemporal context, as well as some discussion as to how this new radar information can inform or validate firn modeling. Those points aside, this manuscript is well-written, complete from an investigative standpoint, and deserving of publication. I recommend minor revisions which I have annotated in the attached .pdf.

Thank you much for your time and effort reviewing our manuscript! We really appreciate the detailed and constructive suggestions, and we agree with all comments and have implemented them in the manuscript. We believe that these changes further improved the manuscript.

Please find answers to all comments below, using the following color code:

Blue indicates comments from the authors

*Green italic indicates changes in the manuscript, with revised text indicated marked as underscored*

**Major Comments:**

Throughout the analysis, I gained a more intimate understanding of how the VHF radar signal interacts with the near-surface, kudos to the authors for this. The efforts put forth in the modeling suggest that a given ice layer in the percolation zone must be buried greater than 8 m as to not interfere with the wave form surface echo. This is approximately 10 years' worth of snow accumulation. However, empirically it was determined and the authors state that "between 2013 and 2014, areas of heterogeneous firn in the dry-snow zone reverted back to homogeneous firn." This disagreement between the modelled information and radar retrieval is not reconciled within the manuscript discussion. Explanation for this phenomenon should be provided.

Thank you for catching this! Together with comments from the other reviewer, we realized that i) the theoretical range resolution we calculated for MCoRDS was too large (due to a wrong $k_t$ value), and ii) the theoretical range resolution for the modeled waveform is likely larger than for the actually measured MCoRDS signal. We re-calculated the theoretical range resolution in firn for the measured MCoRDS data to 5.7 m (instead of 7.5 m), and the range resolution of the modeled signal to ~11 m (calculated using the peak withd at a 3dB drop from the maximum

peak, similar to as the MCoRDS windowing factor for the range resolution is calculated). We attribute the difference between the modeled and observed signal range resolution to the use of different windowing parameters applied in the signal processing, and potentially also due to slightly different input signals.

To explain this, we added the following text to the manuscript: "*We note that the vertical range resolution of the modelled signal, expressed as the pulse width 3 dB down from the peak (similar to the calculation of the MCoRDS windowing factor, https://data.cresis.ku.edu/data/rds/rds_readme.pdf), is ~8.4 m in ice and ~11 m in firn, The larger range resolution is likely due different windowing factors applied in the pulse compression (here we use the standard RadSPy Hanning window), and the input pulse not being an exact replica of the MCoRDS pulse. However, we expect the general behaviour of the modelled signal to be representative of the MCoRDS signal, where depths of layer interfaces should be considered in a relative sense to the different theoretical range resolutions.*"

With this knowledge, we updated Figure 4 (see below) to represent the depth and thickness of ice layers in firn also as a function of the range resolution. From this, it becomes evident that ice layers affect the surface return when in the top 4 m for the observed MCoRDS signal (0.7xRange Resolution), and not the previously stated 8 m. (A similar secondary x-axis was applied to Figure 3, effects of snow depth, to express the thickness as function of range resolution).

[Figure]

*Figure 1: a) Surface peak offset (dz) for RadSPy simulated MCoRDS surface returns over firn stratigraphies consisting of a single ice layer placed at various depths and with different thicknesses. b) and c) show example model input profiles. d) dz for a 0.1 m thick ice layer, as well as layers with densities of 500 kg/m³ and 600 kg/m³ at different depths, with waveforms for selected ice layer depths (marked with crosses) shown in e and f. g) dz for an ice layer, as well as layers with densities of 500 kg/m³ and 600 kg/m³, starting at 1 m depth and with increasing layer thickness, with waveforms for selected ice layer thicknesses (marked with crosses) show in h and i. The dotted lines in the waveform plots show the picked peak identified used to calculate dz. The x-axis on the top of panels d) and g) show the ice layer depth and thickness scaled by the rang resolution δ_r of the modelled signal in firn (~11 m).*

We changed the text in the results section of the manuscript accordingly:

*"When an ice layer lies at depths greater than ~0.7δ_r, thus ~4 m for the MCoRDS signal in firn, the radar's surface return displays two distinct peaks.... Thus, ice layers located deeper than ~0.7δ_r have negligible impact on the surface reflection peaks, leading to dz ≈ 0."*

This now matches better with observations where heterogeneous firn switches from 2013 to homogeneous firn in 2014, where burying the 2012 melt layer to 4 m depth between 2012-2014 is more reasonable. We also note that the change in *dz* only represents a "bulk" near-surface heterogeneity, and not a heterogeneity at a specific depth. For example, *dz* could have been most sensitive to already deeper layers, which then were pushed out of the sensitive depth (~4-5.7 m) in the following year. Thus, not the entire 4-5.7 m firn column needs to be replaced with homogeneous firn. To clarify this, we added the following sentence in the discussion section:

*"We also note that dz represents a bulk near-surface heterogeneity over the MCoRDS surface return sensitive depth and cannot be used to identify heterogeneity/ice layers at a specific depth."*

The correct identification of the "true" radar surface required an algorithm which upsamples the MCoRDS data to finer fast-time resolution. The author's choice of resampling algorithm has a significant effect on the outcome of their results. If for example a piecewise linear upsampling was applied, the original MCoRDS signal and the upsampled signal would appear identical. Using a Fourier method as you have described, introduces "Gibb's Phenomenon" which can be seen as the oscillations at the tail end of the surface reflection signal. I suspect also that such phenomena are occurring to produce the troughs in the signal seen in Figure 6d & e (i.e., the Fourier series is struggling to represent the flat discontinuity of the signal between samples 3&4 of 6d). I appreciate the approach you have taken, and it is fortuitous that the upsampled signal reasonably recreates the modelled signal. However, acknowledging the origins of this "spurious" waveform bulge, be that an effect of the resampling algorithm or your choice in sample lags, should be considered and described more thoroughly.

Thank you for this comment. We would like to note that the trough in the signal is not "spurious", but is a real signal (as suggested from the modeling), which we attribute to the interaction between the radar signal and the heterogeneous subsurface. We argue that the observed MCoRDS signal simply does no show such troughs because of the lower sampling rate, therefore not capturing the complete signal. Thus, we intentionally upsample the signal with a Fourier transform introducing the Gibbs Phenomenon, allowing us to recover the first return (i.e. getting a peak and trough before the maximum peak).

We tried to make this more clear in the manuscript by adding: "*Therefore, using an upsampling approach based on a Fourier method that introduces oscillations that recovers closely spaced waveforms is a reasonable strategy for picking the first return. However, we note that depending on the relative position of the signal peak, this phenomenon can cause some uncertainties in dz.*"

We also added a sentence in the discussion section outlining that changes in dz could occur from the location of the signal peak relative to the waveform sampling:

"*While our modelling and observations generally align, some discrepancies exist. For instance, modelling predicts dz = 0 over the ablation- and dry-snow zones, yet our observations also include non-zero (and negative) values in these areas. Such deviations could stem from factors not accounted for in the model, such as the surface slope (given that the laser is nadir looking, versus the radar records the nearest return, which may result in negative dz values over sloping surfaces) surface roughness and surface anomalies like crevasses. Additionally, uncertainties in dz may arise from how the MCoRDS signal is sampled,….*"

Explanations of why dz values near the summit of Greenland are significantly higher than the surrounding dry-snow zone data retrievals remains unsubstantiated. Answers to these concerns are not supported in Scanlan et al. (2023), which pertain to higher frequency radar systems and show both consistently low retrieved density and high density within the same region of Greenland's summit. Through tighter analysis, or a fleshed-out hypothesis, effort should be considered to reconcile the work of these authors.

We agree that the discussion on these anomalies could have been a bit more thorough. We changed this paragraph to include the possibility of small density contrasts from thin layers and

surface roughness. Also changed the wording to make the connection to Scanlan et al (2023) - we mean to point out the similarity of these unexpected observations rather than explaining our results with the high-densities found in Scanlan et al (2023).

*"Our results also reveal areas with complex surface processes, for example, localised high dz areas in the dry-snow zone, particularly near the summit of the ice sheet. It is possible that these areas consist of a high small-scale density variability in the near-surface (e.g. from wind scour or hoar formation) (Hörhold et al., 2011), causing strong enough density contrasts to affect the radar surface peak. Additionally, increased dz may be caused by increased surface roughness (e.g. from sastrugi), which introduces waveform interferences when the radar signal from facets with different heights when the signal is coherently integrated during processing. We note that this observation aligns with unexpectedly high surface densities in the summit region reported by Scanlan et al., (2023). The persistence of this anomalous feature over several years is unclear, but may be attributed to the relatively low snowfall in the area, implying that the near-surface varies only over long timescales."*

**Minor Comments:** We addressed the minor comments from the PDF as follows:

L15: *"In this study, we use concurrent VHF airborne radar (MCoRDS, 195 MHz) radar and laser altimetry (ATM)…"*

L16: *"...to investigate our hypothesis that heterogeneities in firn (i.e. ice layers) cause vertical offsets in the radar surface reflection (dz).*

L18: *"... effectively delineates between vertically homogeneous and vertically heterogeneous firn profiles over a depth range of ~4 m"*

L25: Changed *survived* to *endured*

L30: Done

L32: Done

L95: We added the typical width of the surface return (using the examples from Figure 2): *"... (hereby referred to as the broader surface signal of elevated amplitudes, typically 0.1-0.3 µs wide, and can encompass multiple peaks)…"*

L125: Done

L138: Done

L139-142: We agree that this section includes some observations from the modeling results section, but we used these to help calibrate the radar peak offsets. We therefore keep this part here to help explain how we identified systematic offsets. However, we added a sentence about potential remaining timing issues in the radar measurements to the discussion section:

*"Such deviations could stem from factors not accounted for in the model, such as the surface slope (given that the laser is nadir looking, versus the radar records the nearest return, which may result in negative dz values over sloping surfaces) surface roughness and surface anomalies like crevasses. Additionally, uncertainties in dz may arise from how the MCoRDS signal is sampled, and timing issues (i.e. cable delays) in the radar measurements that were not fully identified and corrected in our calibration."*

L142: Done

L143: We split the sentences and change it to: *"The calibration zone started 20 km East from the western ice margin to avoid the steepest and most crevassed part of the ice, and extended to…"*

L145: Done

L162: Done

L164-165: We changed the sentence to: *"Additionally, we added a snow layer of varying thickness (up to 2 m) to the ablation zone model (three-layer model), using a density of 341 kg/$m^3$…"*

L256: Done

L273-275: We agree that there is a bit of methods in this paragraph, but we believe it is easier to follow if this stays in the "Effects of waveform sampling" section and therefore leave it here.

L287: addressed above

L290: This is explained in the first paragraph of this section. To help make this connection, we replace the term "*lag*" with "*varying time-zero offsets*" to be consistent.

L297: That is a good point with the absolute values, and we changed it to: *"Specifically, low absolute dz values (<1m)…"*

L375: Done

L385: Done

424: Curious as to what the correlation between dz and MAR melt is. Correlation would likely break down in the ablation zone, but it would be interesting/confirmatory data in the percolation and dry zones.

We agree that it could be interesting to do a correlation between dz and MAR melt. However, we do not include such an analysis here as we believe the manuscript contains enough information to show the spatial correlation between dz and the percolation zone, which was derived from the mean MAR melt.

L471: What is the proposed cause of higher firn densities near the summit?

We agree that the discussion on these anomalies could have been a bit more thorough. We changed this paragraph to include the possibility of thin higher-density layers or surface roughness.

*"Our results also reveal areas with complex surface processes, for example, localised high dz areas in the dry-snow zone, particularly near the summit of the ice sheet. It is possible that these areas consist of a high small-scale density variability in the near-surface (e.g. from wind scour or hoar formation) (Hörhold et al., 2011), causing strong enough density contrasts to affect the radar surface peak. Additionally, increased dz may be caused by increased surface roughness (e.g. from sastrugi), which introduces waveform interferences when the radar signal from facets with different heights when the signal is coherently integrated during processing. We note that this observation aligns with unexpectedly high surface densities in the summit*

region *reported by Scanlan et al., (2023). The persistence of this anomalous feature over several years is unclear, but may be attributed to the relatively low snowfall in the area, implying that the near-surface varies only over long timescales."*

L498: We agree that giving a vertical depth is useful – following the modeling results, we use the depth of 0.7 x the theoretical vertical range resolution as a maximum depth, and change this in the manuscript to: *"Our results, supported by modelling and in-situ firn core analyses, demonstrate that dz serves as an effective tool to delineate between vertically homogeneous and heterogeneous firn profiles ~0-4 m depth, where temporal changes in dz over 2-5 years align well with known climatic events."*

L503: Done

L505: Done

**References added:**

Hörhold, M. W., Kipfstuhl, S., Wilhelms, F., Freitag, J., and Frenzel, A.: The densification of layered polar firn, Journal of Geophysical Research: Earth Surface, 116, https://doi.org/10.1029/2009JF001630, 2011.